# Higher order genetic interactions switch cancer genes from two-hit to one-hit drivers

Solip Park [1✉], Fran Supek [2,3✉] & Ben Lehner [3,4,5✉]

The classic two-hit model posits that both alleles of a tumor suppressor gene (TSG) must be inactivated to cause cancer. In contrast, for some oncogenes and haploinsufficient TSGs, a single genetic alteration can suffice to increase tumor fitness. Here, by quantifying the interactions between mutations and copy number alterations (CNAs) across 10,000 tumors, we show that many cancer genes actually switch between acting as one-hit or two-hit drivers. Third order genetic interactions identify the causes of some of these switches in dominance and dosage sensitivity as mutations in other genes in the same biological pathway. The correct genetic model for a gene thus depends on the other mutations in a genome, with a second hit in the same gene or an alteration in a different gene in the same pathway sometimes representing alternative evolutionary paths to cancer.

[1] Centro Nacional de Investigaciones Oncológicas (CNIO), Madrid, Spain. [2] Institute for Research in Biomedicine (IRB Barcelona), The Barcelona Institute of Science and Technology (BIST), Barcelona, Spain. [3] Catalan Institution for Research and Advanced Studies (ICREA), Barcelona, Spain. [4] Centre for Genomic Regulation (CRG), The Barcelona Institute of Science and Technology (BIST), Barcelona, Spain. [5] Universitat Pompeu Fabra (UPF), Barcelona, Spain. ✉email: solippark@cnio.es; fran.supek@irbbarcelona.org; ben.lehner@crg.eu

Cancer driver genes are classified as oncogenes (OGs) or tumor suppressor genes (TSGs) depending upon whether their activation or inactivation contributes to cancer. Whereas a single mutation in an OG can be sufficient to increase tumor fitness, inactivation of both copies of a TSG is often required, as envisaged by the classic two-hit hypothesis[1,2]. However, >500 genes have now been causally implicated in cancer[3] and exceptions to these models are known[4,5]. For example, some OGs are dosage-sensitive, with amplification of a mutated copy further increasing tumor fitness[6–8] and some TSGs are haploinsufficient, with inactivation of a single allele promoting cancer[4,9–11].

Large-scale cancer genome sequencing presents an opportunity to systematically investigate the dosage sensitivity of OGs and the dominance of TSGs and the extent to which these are fixed or variable. In model organisms such as yeast, the activity-fitness functions of genes have been systematically experimentally quantified. For growth rate, activity-fitness functions are typically non-linear, are frequently 'peaked' (i.e., non-monotonic with maximal fitness at an intermediate activity level), and can change across conditions[12]. Consistent with this, whether some TSGs behave as haploinsufficient one-hit drivers or recessive two-hit drivers has been reported to vary across cancer types and patients[4,13–15] and mutated OGs can be highly amplified in one cancer type but not in others[16].

Using data from ~10,000 tumors we show here that both OGs and TSGs quite often vary in whether they behave as one-hit or two-hit drivers. These changes in dosage sensitivity and dominance are examples of the interactions between mutations being contingent upon the context. In model organisms, such changes are often caused by higher-order epistasis with, for example, a third mutation modifying the interaction between two alterations[17–20]. We find that higher-order interactions are also important in human tumors and use third-order genetic interactions to identify mutations that switch cancer genes between behaving as two-hit and one-hit drivers. Taken together, our results suggest that the second hit in one driver and a hit in another driver in the same biological pathway can sometimes have similar consequences and be alternative evolutionary paths to cancer.

## Results

**Interactions between mutations and CNA in 10,000 tumors.** We employed a statistical test based on log-linear regression - a generalization of the chi-square test to more than two dimensions (see Methods; Fig. 1a, b)—to identify interactions between somatic mutations and copy number alterations (CNAs) in 201 cancer driver genes (see Methods) across ~10,000 tumors representing 33 types of cancer characterized as part of the TCGA project[21–23]. We tested for co-occurrence between mutations and either CNA gain (or amplification) or CNA loss for each gene in the cancer types in which it is significantly mutated (>2% mutation frequency in a single cancer type; a mean of 2.3 (median = 1) cancer types per gene, 454 gene-cancer type pairs in total. The 201 genes include 117 TSGs, 77 OGs, and 7 dual-functional genes (DFGs) (genes classed as both TSGs and OGs in different cancer types) (see Methods)[21].

In total, interactions between mutations and CNAs were detected for 40 genes (19.9% of the tested genes) in at least one cancer type and for 17.4% of all tested gene-cancer type pairs (false discovery rate, FDR = 10%; Supplementary Figs. 1 and 2a). The 40 genes had interactions between mutations and CNAs in a mean of 2.2 cancer types (median = 1, range: 1–17, with TP53 having the most interactions). Interactions were detected for 26 TSGs (65.0% of the detected genes), 12 OGs (30.0 %) with a total of 63 interactions between mutations and CNA loss, and 24 interactions between mutations and CNA gain (8 interactions were detected in both the

loss and gain models; FDR = 10%, Fig. 1c; the results for all tested pairs are shown in Supplementary Fig. 3 and Supplementary Data 1). Consistent with the two-hit model, 56/63 interactions between mutations and CNA loss (88.9%) were for TSGs and 15/24 interactions between mutations and CNA gain (62.5%) were for OGs. At FDR 20%, interactions between mutations and CNAs were detected for 56 genes (27.9% of the tested genes) in at least one cancer type and for 23.6% of all tested gene-cancer type pairs.

**Four classes of driver genes.** Clustering suggested the 73 genes tested for interactions between mutation and CNAs in at least two types of cancer fall into four major classes (Fig. 1c; Supplementary Fig. 2b).

First, 23 TSGs, 18 OGs, and 3 other genes showed patterns consistent with them primarily functioning as one-hit drivers with no significant co-occurrence between mutations and CNAs in any cancer types in which they are significantly mutated (FDR = 10%). Examples of class I ('one-hit') drivers include the genes SF3B1, NOTCH1, and IDH1.

Second, 15 TSGs, 1 OG, and 2 DFGs only had interactions between mutations and CNA loss in at least one cancer type, consistent with them acting in at least some cancers as two-hit drivers. These class 2 ('two-hit loss') drivers include the genes RB1 (3/13 cancer types), NF1 (5/8), NF2 (1/2), PTEN (7/15), and BAP1 (2/6).

Third, 2 TSGs and 6 OGs only had interactions between mutations and CNA gain. These class 3 ('two-hit gain') drivers include EGFR (2/3 cancer types), KRAS (4/16), and BRAF (1/6). Changes in VAFs suggest it is the mutant allele that is normally amplified for class 3 genes (Supplementary Fig. 5).

Fourth, a set of 3 cancer genes had interactions between mutations and both CNA loss and gain. These class 4 ('two-hit loss and gain') drivers include 1 TSG where mutations interact with CNA gain in one cancer type but with CNA loss in a different type of cancer and 1 TSG and 1 OG where mutations interact with both CNA gain and loss in the same cancer type. For example, mutations in CUL3 interact with CNA loss in one type of cancer (kidney renal papillary cell carcinoma) but with CNA gain in head and neck squamous cell carcinomas and with no interactions detected in one additional cancer type in with the gene is significantly mutated. An additional striking example is TP53 which has interactions between mutations and CNA loss in 16 cancer types, between mutations and CNA gain in 7 cancer types, and between mutations and both CNA loss and gain in 6 cancer types.

We also investigated the alternative possibilities of two-way interactions through (i) promoter DNA hypermethylation (silencing) and somatic mutation or (ii) promoter DNA hypermethylation and CNA loss. Using 31 gene-tissue pairs in which a cancer gene is epigenetically silenced in >1% of samples, only one significant two-way interaction between promoter DNA hypermethylation and CNA loss–ZNF133 in ovarian cancer—was identified (FDR 10%; Supplementary Fig. 7b; Supplementary Data 2).

**Changes in the interactions between mutations and CNAs.** Although the class 4 drivers are extreme examples of the interactions between mutations and CNAs changing across contexts (cancer types), the data suggest that this is also true for many of the other drivers (Fig. 1c). For example, whereas mutations in the classic TSG NF1 interact with CNA loss in most cancer types (62.5%), the driver mutations in BRAF only interact with CNA gain in one of the four cancer types in which it is significantly mutated (skin cutaneous melanoma; SKCM) (Fig. 2a, Supplementary Fig. 6). To further explore changes in these interactions, we tested whether the strength of interaction between mutations and CNAs differs between detected cancer type (cancer type-specific two-way interaction; FDR = 10%)

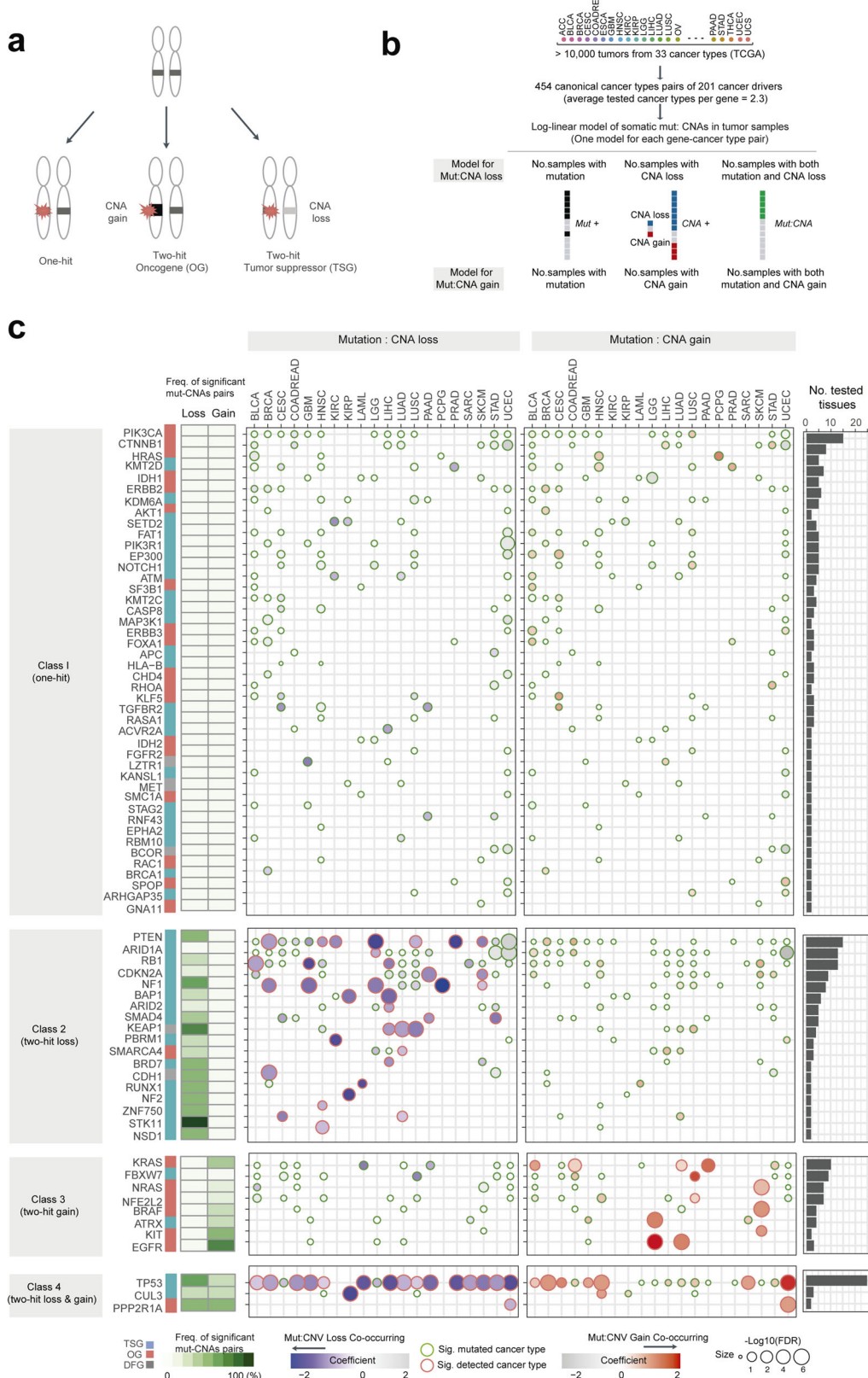

**Fig. 1 Interactions between somatic mutations and CNAs in human tumors. a** Classical one-hit and two-hit models. **b** Log-linear regression for identifying interactions between mutations and CNAs. **c** Interaction coefficient (i.e., effect size) and FDR-values in each cancer type for 73 cancer genes tested in at least two cancer types. Cancer types with >150 samples and at least 1 significant interaction are shown (FDR = 10%). The full data set is shown in Supplementary Fig. 3. Cancer-type abbreviations are listed in Supplementary Table 1. Source data underlying **c** are provided as a Source Data file.

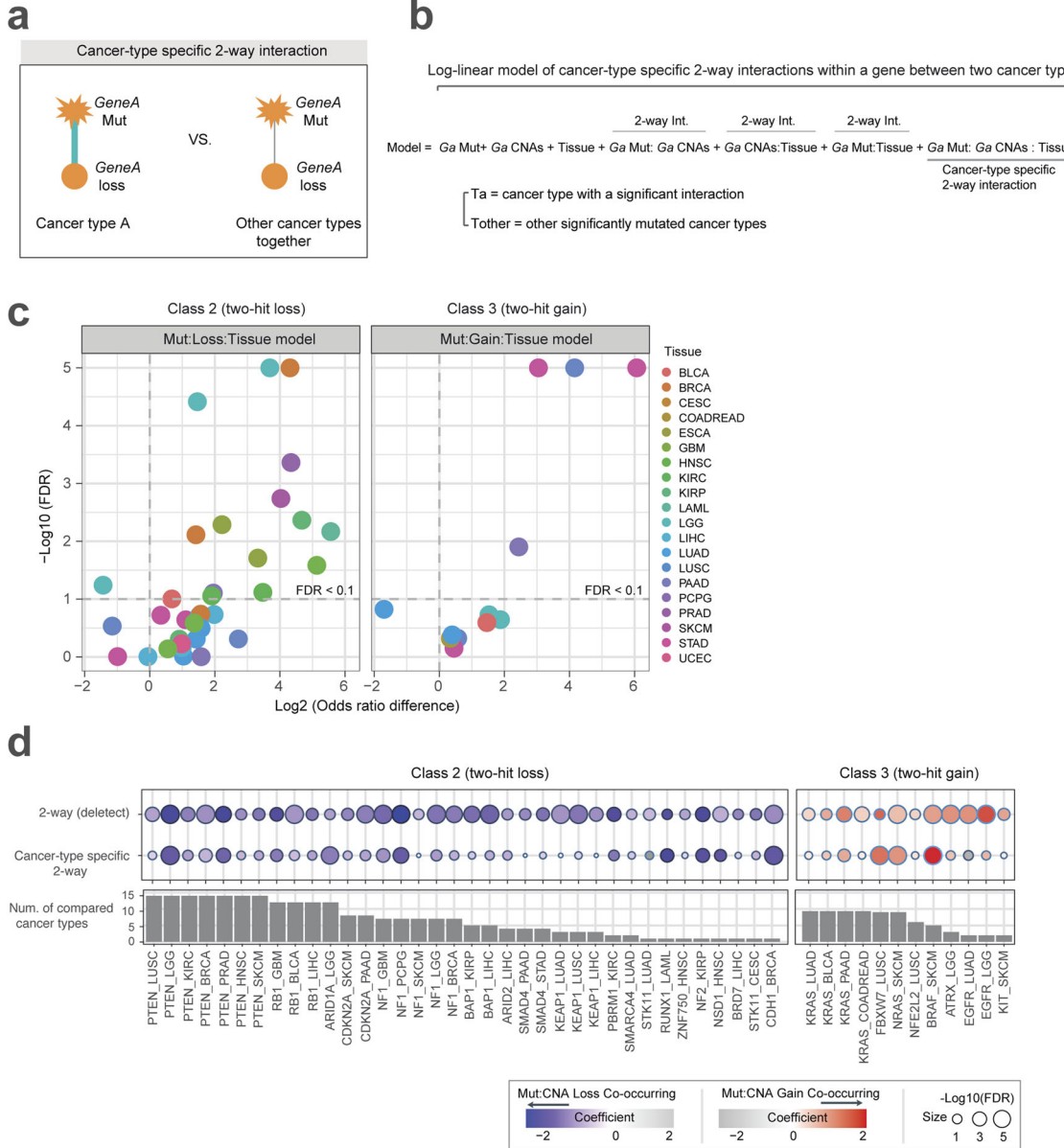

**Fig. 2 Interactions between mutations and CNAs change across cancer types. a** Tested hypothesis. **b** Test for whether interactions change between cancer type A (detected; FDR 10%) and other cancers in which the gene is significantly mutated (compared). *Ga* Mut denotes the number of samples with somatic mutation of gene A in cancer type with a significant interaction, *Ga* CNAs indicates a number of samples with CNAs of gene A in cancer type with a significant interaction, and *Tissue* denotes the number of samples with genomic alterations of gene A in other significantly mutated cancer types. **c** Volcano plot comparing differences of the log of the odds ratios for the co-occurrences between mutation and CNA in two cancer types (i.e., detected cancer type and other significantly mutated cancer types together). A total of 48 detected interactions for 26 genes were tested. Color coding is for the cancer type in which the two-way interaction was detected (FDR = 10%). 0.5 was added to each frequency when calculating odds ratios to avoid division by zero frequencies. **d** Effect sizes (interaction coefficients) and FDR values for tissue-specific interactions were estimated as the ratio between the number of detected interactions in the permutated matrix and the number of detected with the real data with 100 permutations. Source data underlying **c** and **d** are provided as a Source Data file.

and other cancers in which the gene is mutated (>2%) using log-linear regression (Fig. 2b; Supplementary Data 3). We tested whether 48 interactions for 26 genes (36 interactions for 18 genes in class 2 and 12 interactions for 8 genes in class 3, FDR = 10% in one cancer type) changed in strength in other significantly mutated cancers together (median = 3 cancer types; mean = 4.5) (Fig. 2c, d and Supplementary Fig. 6). This analysis revealed that 41.6% (20/48) of the interactions differ in strength between cancer types (FDR = 10%; 16 interactions from 10 genes in class 2, 4 interactions from 4 genes in class 3). For example, the interaction between *BRAF* mutation and CNA gain in SKCM is stronger than in the three other types in which *BRAF* is significantly mutated (FDR = 10%).

**Third-order interactions switch genes from two-hit to one-hit drivers.** Why do the interactions between mutations and CNAs change in different cancer types and even within the same type of cancer? One cause of changes in the pairwise interactions between mutations in model organisms is higher-order genetic interactions (also called higher-order epistasis)[18,24,25]. In the simplest examples—third-order interactions—the interaction between two mutations changes depending upon whether a third mutation is present or not. Third-order interactions can occur among mutations in the same[19,26,27] or different[17,28] genes and make important contributions to diverse phenotypic traits such as growth and drug resistance[18,29]. We hypothesized that conceptually similar higher-order interactions

might be occurring in cancer genomes, with mutations in a second gene (between gene interaction) altering the interaction between a mutation and CNA in a cancer gene (within gene interaction) (Fig. 3a). Specifically, we tested for third-order interactions involving two genetic alterations in one gene (somatic mutation and CNA) and the third alteration in a second gene (somatic mutation). To identify third-order genetic interactions, we only considered somatic mutations in the second gene, not including copy-number changes in the second gene to avoid possible confounding by the overall level of copy-number variation. Since many tumors carry more than two driver mutations[30], there is plenty of opportunity for higher-order interactions amongst driver mutations.

We used log-linear regression to test for third-order interactions between mutations and CNAs in one gene (gene A) and mutations in a second gene (gene B, Fig. 3b). Across cancer types, we were able to test for third-order interactions for 40 genes with 79 pairwise interactions between mutations and CNAs (63 interactions for 30 genes with CNA loss and 24 for 14 genes with CNA gain; 4 genes with both). Each pairwise interaction was tested for interactions with mutations in a mean of 19.1 other driver genes (median = 19) mutated in at least 2% of samples, with a total of 1511 third-order interactions tested (Supplementary Data 4).

In total, we identified 17 third-order interactions (FDR = 10%, this likely underestimates the true number of higher-order interactions because of the low statistical power to detect them). To illustrate how the presence of mutations in a second driver gene (gene B) alters the interaction between mutations and CNAs in a first driver (gene A), in Fig. 3c–f, we divide samples according to whether they do (Fig. 3d) or do not (Fig.3e) carry mutations in gene B and then plot the frequency of gene A CNA in samples carrying gene A mutations. Most of the third-order interactions (76.5%, 13/17) are examples where the presence of a mutation in a second driver gene decreases the strength of the interaction between mutations and CNAs in the first driver (Fig. 3c–f). For example, there is a strong interaction between mutations and CNA loss in KEAP1 in lung squamous cell carcinoma (LUSC) but not in samples that also carry a PTEN mutation. This suggests that mutations in PTEN sensitize lung cells to the effects of reduced KEAP1 activity. Similarly, in Fig. 3g we show how the frequency of gene A mutations in samples carrying gene A CNAs varies depending upon the presence of mutations in gene B. For example, there is a strong third-order interaction between BRAF mutations, BRAF CNA gain, and NRAS mutations in SKCM. Only 2.7% of samples with BRAF CNA gain and NRAS mutations have BRAF mutations whereas 81.9% of samples with BRAF CNA gain without NRAS mutations have BRAF mutations. This is consistent with mutations in both genes activating the same pathway.

**Second hits in the same pathway switch genes from two-hit to one-hit drivers**. Strikingly, most of the third-order interactions involve two genes from the same canonical cancer signaling pathway[31]. This includes the PI3K pathway (BRCA and UCEC), RTK/RAS pathway (SKCM), Nrf2 pathway (LUSC), TBF-β/SMAD4 pathway (COADREAD), cell growth pathway (LGG and LIHC), and p53 pathway (LUAD and LGG). Genes participating in third-order interactions are indeed enriched for shared functions and pathway membership (Fig. 3h; Fig. 4).

This suggests a simple principle for why these third-order interactions occur and why cancer genes switch between being one-hit and two-hit drivers: two hits in one gene in a pathway or two hits in two different genes in a pathway can have similar functional consequences and so act as an alternative (partially redundant) evolutionary paths during tumor progression. Using this principle, one additional third-order interaction (RB1-ASXL2 in BLCA) could be identified when gene pairs were restricted to

genes sharing at least one Gene Ontology molecular process or pathway annotation (FDR = 10%; Supplementary Data 5).

## Discussion

We have shown here in an analysis of ~10,000 tumors that whether a cancer gene requires only one or two genetic alterations to contribute to cancer typically varies across different types of cancer and across individuals. We have also shown that higher-order genetic interactions are important in human tumors, i.e., that in order to understand the genetics of cancer, not only do the effects of individual mutations and their pairwise interactions need to be considered, but also what happens when three or more alterations are combined. Indeed, we have identified multiple examples where the pairwise interaction between two alterations changes when a third alteration is present in a cancer genome.

Quantifying these third-order interactions allows us to propose a simple principle for tumor evolution (Fig. 5): that for some cancer types (e.g., SKCM) tumor evolution can occur via two alternative evolutionary paths—either via a cell obtaining two hits in a single driver gene (e.g., BRAF) or via a cell obtaining single hits in two different genes in the same pathway (e.g., BRAF and NRAS). Put another way, a pathway needs two genetic alterations to be (in) activated, but these alterations can either both be in the same gene or they can be in two different genes in the pathway. This mutual exclusivity is observed in multiple cancer pathways and for multiple types of cancer (Fig. 3h; Fig. 4). Indeed, this might reflect a more general principle of genetic architecture that a strong perturbation in one gene can have a similar functional consequence as the combination of two weaker perturbations in two different genes in a pathway.

In contrast to the changing activity-fitness functions of many drivers, a subset of cancer genes (class 1 drivers) appears to nearly always behave as one-hit drivers, consistent with second alterations conferring either no benefit or actually a fitness cost to a tumor. Consistent with this second possibility, many class 1 TSGs have been previously identified as essential genes (Supplementary Table 2), suggesting these TSGs may have 'peaked' (non-monotonic) activity-fitness functions whereby reduced activity promotes cancer but complete inactivation is lethal to a cell. Two-hits in these drivers may never (or only very rarely) be compatible with cell viability. This suggests the intriguing translational implication that either re-activating or further inactivating class 1 TSGs may be viable therapeutic strategies to pursue.

Finally, we note that, although the results presented here apply to human cancers, it is likely that the principles about genetic architecture will also apply to other diseases and traits. The demonstration of the importance of higher-order interactions in cancer suggests that they will also contribute to the genetic architecture of other complex diseases. Moreover, it seems likely that additional diseases beyond cancer will also have alternative 'within' and 'between' locus causes, with disease resulting from the combination of mutations in one gene in some individuals and from the interaction between mutations in two different genes in others. More generally, similar principles may apply during evolution, with multiple mutations in a single gene and the interactions between mutations in different genes in a pathway representing parallel paths to the evolution of new phenotypic traits.

## Methods

**Sample preparation**. We used comprehensive molecular datasets collected across 33 cancer types from 11,276 patients by the TCGA project[23]. We only considered samples that had available data across two genomic platforms: somatic mutations and CNAs. To obtain high-quality data set, we discarded samples that have been flagged with quality control issues or during pathology review (merged_sample_-quality_annotations.tsv). After applying these filtered, hypermutated samples were also excluded (more variant than third quartile + interquartile range × 1.5). A total of 9175 patients' data were used in this work.

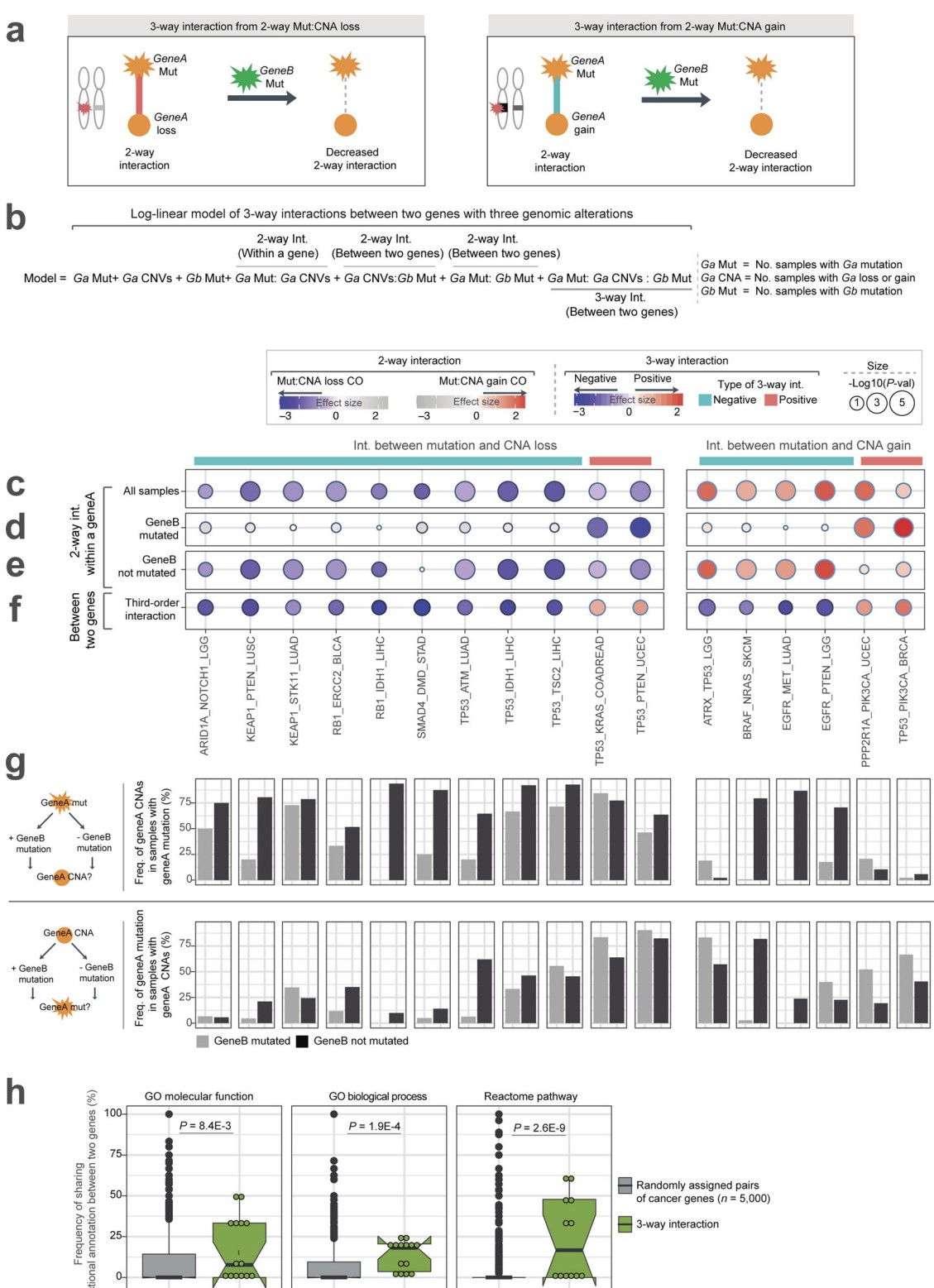

**Somatic genomic alteration.** Genomic data from TCGA Data were obtained from TCGA data Portal (Pan-Can Atlas). We downloaded version 2.8 of the mutation annotation format (MAF) file provided by the 'Multi-Center Mutation Calling in Multiple Cancers' (MC3) project, as a part of the TCGA Pan-Cancer Atlas effort[22]. These unified MC3 somatic mutations were called from seven software packages using a single-standardized protocol across many different individual studies (mc3.v0.2.8.PUBLIC.maf.gz). It provides high-quality variant calls after applying rigorous filtering steps to discard low-quality variants and remove possible sequencing artifacts. Somatic mutation calls were assigned to all premature truncation mutations (encompassing splicing variants, frameshift indels, and nonsense variants) and to non-synonymous (missense mutation), single-residue substitutions (in-frame indels). Predicted deleterious missense was assigned at least one of two tools (SIFT and PolyPhen2) predicted as deleterious/damaging variant[32,33]. Genomic regions with significant levels of CN arrangements and their target genes of these somatic CNAs were determined using GISTIC 2.0 with their $q$-values[34]. Gene-level CN data were obtained from Synapse (syn5049520). High-level deletion for a gene was defined as GISTIC threshold CN value of $-2$, whereas high-level amplification for a gene was assigned with threshold CN value $+2$.

**Fig. 3 Higher-order genetic interactions identify mutations that switch genes from two-hit to one-hit drivers. a** A third-order genetic interaction means the strength of a pairwise interaction (between mutation and CNA in the first gene) changes when a second gene is mutated. **b** Log-linear regression for identifying third-order interactions between mutations and CNAs in one gene and mutations in a second gene. The three-way model quantifies the strength of interactions of two genomic alterations within a single gene with a background alteration of the second gene (gene B) mutation in a cancer type. *Ga* Mut denotes the number of samples with somatic mutation of gene A, *Ga* CNAs indicates a number of samples with CNAs of gene A, and *Gb* Mut denotes the number of samples with somatic mutation of gene B. **c–f** Effect sizes (interaction coefficients) and two-sided *P*-values for 17 third-order interactions. Interactions between mutation and CNA in all samples (**c**), samples with gene B mutations (**d**), and samples without gene B mutations (**e**), and third-order interactions (**f**). **g** Frequencies of gene A CNAs in samples with gene A mutations (top panel) or gene A mutations in samples with gene A CNAs (bottom panel) depending on whether gene B is mutated or not. **h** Gene pairs involved in third-order interactions share functions and pathways more than random pairs of cancer genes (*P*-values from two-sided Mann–Whitney *U* test). The median value of each gene set is displayed as a band inside each box. The length of each whisker is 1.5 times the interquartile range (shown as the height of each box). Source data underlying **c–h** are provided as a Source Data file.

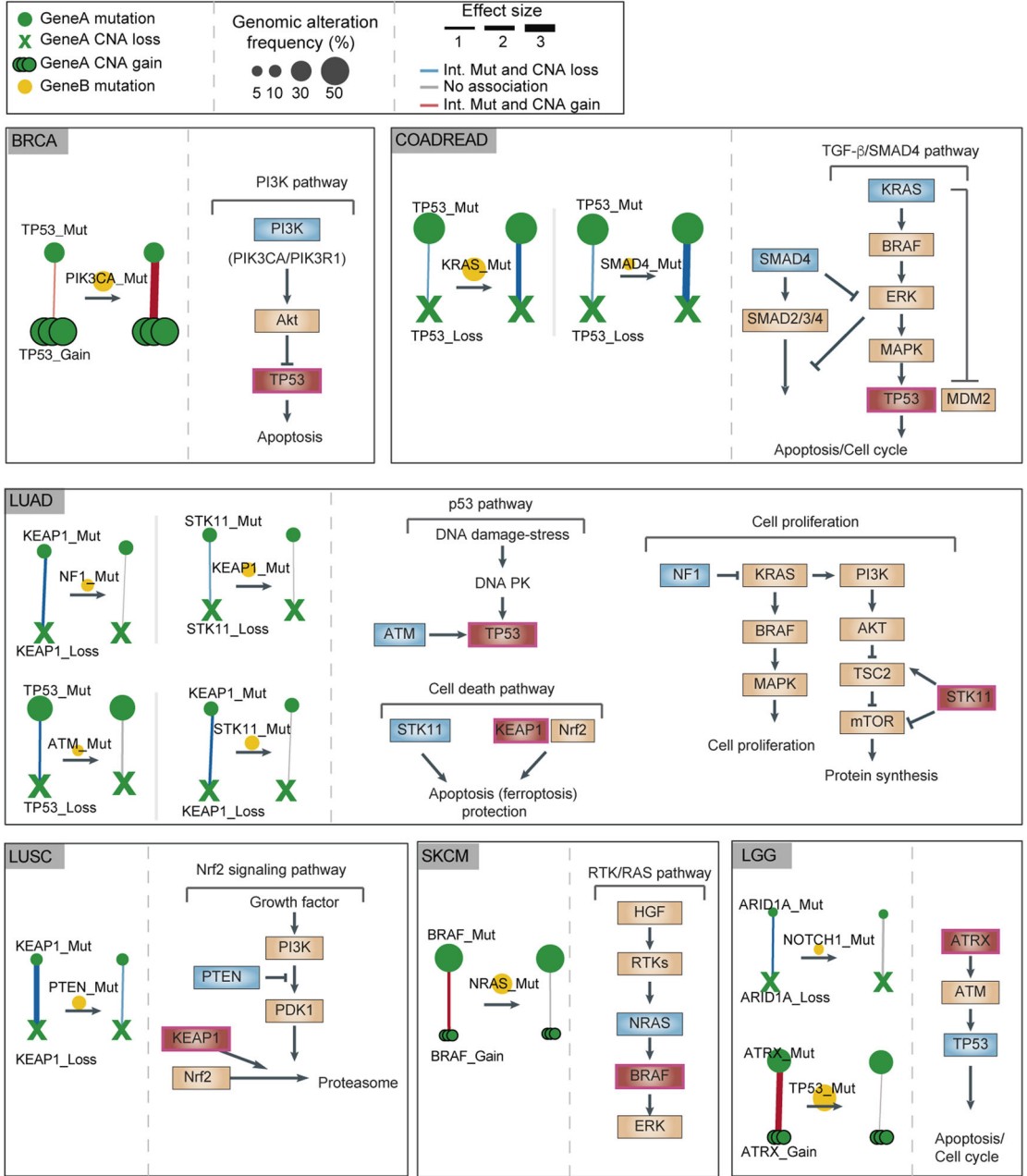

**Fig. 4 Mutations that switch genes from two-hit to one-hit drivers in six cancer types.** Changes in the strength of interaction between mutations and CNAs in driver genes ('gene A') in the absence or presence of mutations in a second cancer gene ('gene B'). A summary of the pathway in which the function of the gene is also shown.

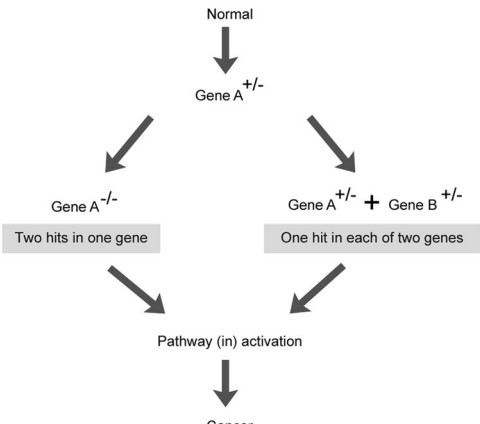

**Fig. 5 Alternative evolutionary paths to cancer.** Once a driver gene is mutated, either a second hit in the same gene or an alteration in a different gene in the same biological pathway can be alternative evolutionary paths to cancer. That is, for some pathways, two hits may be required to (in) activate it but these hits can either be in one gene or in two different genes. The ordering of events may differ from that illustrated here.

Broad-level deletion (loss) and broad-level amplification (gain) were estimated by GISTIC threshold CN values having less than −1 or greater than +1, respectively.

**Somatic driver set.** To characterize interactions between mutations and CNAs, we compiled a high-confidence list of 201 somatic cancer driver genes. First, 235 cancer genes were collected by the union of genes predicted by the eight driver gene predictors (20/20+, ActiveDriver, CompositeDriver, MuSiC, MutSig2CV, OncodriveCLUST, OncodriveFML, and e-Driver), manually reference search, and individual TCGA studies from a Pan-Cancer Atlas[21]. The entire process is described in more detail in[21]. Next, cancer genes were selected if they were: (1) listed as 'oncogene' or 'tumor suppressor' at least one single cancer type (not in 'PANCAN') and (2) mutation frequencies in a certain cancer type >2%.

Collected somatic drivers were categorized either as TSG or OG according to 20/20+ predictors across each cancer types[35]. DFGs are genes classified as both TSGs and OGs in different cancer types. This method is based on an improved version of the 20/20 rule (gene is considered to be a TSG when a gene has >20% truncating mutations, whereas OG, will be defined with >20% missense mutations)[36] using Random Forest machine learning algorithm for classifying TSG and OG from somatic mutations. It applies five different features: capturing mutational clustering, evolutionary conservation, predicted functional impact of variants, mutation consequence types, gene interaction network connectivity, and other relevant covariates. In addition, genes in the cancer types in which it is significantly mutated (>2% mutation frequency in a single cancer type) were only considered. Finally, 281 TSGs-single tissue pairs and 173 OGs-single tissue pairs were assigned from 201 genes (55 genes were not included they were not assigned any category either TSG or OG). TSGs and their cancer types were defined when they were annotated as 'tsg' or 'possible tsg', whereas OGs and their types were assigned with the annotation as 'oncogene' or 'possible oncogene'.

**Alternative epi/genomic alterations.** Multiple driver mutations (MMs) in cancer genes were obtained from Saito et al. using DNVChecker in five cancer cohorts, of which 9230 TCGA samples[37]. It is designed to detect multiple single-nucleotide variants observed in the same codon and also check their allelic frequencies with the corresponding BAM files to define cis or trans-MMs.

Epigenetic silencing (promoter DNA hypermethylation) events of cancer genes in TCGA samples were obtained from Saghafinia et al. using the RESET method[38]. After collecting only probes mapping to a gene promoter region, high DNA methylation probes were defined with mean $\beta$-values (0, minimal level of DNA methylation; 1, the maximal level of DNA methylation) higher than 0.8 and standard deviation lower than 0.005 in normal samples. Next, the association between DNA hypermethylation (probe $p$) and a significant decrease of mRNA expression (gene $g$ that corresponds to $p$) was analyzed to evaluate the effect of aberrant DNA methylation states. After 100 randomizations to test the significance, a hypermethylation call with FDR < 0.1 was selected. For running a log-linear regression model for two-way interaction between hypermethylation and CNA loss (or mutation), we converted hypermethylation events from probes to genes when >50% of corresponding probes have hypermethylation events.

Allelic imbalance (AI) was determined by testing for a change in variant allele frequencies (VAFs) in the tumor sample compared to in the matched non-tumor sample (mainly from blood) for each patient from our previous study[39]. To define AI in each sample across genes, we first collected all germline variants in the coding and noncoding regions within each gene (expanding the analyzed region

bidirectionally to 100 kb for genes shorter than this size to reduce the gene length bias). Next, a two-tailed Fisher's test that compares the sequencing read counts collecting variants and the reference alleles in the tumor were performed with each collected variant in a gene. The $P$-values from all collected variants in a gene were then pooled by Fisher's method for combing $P$-values. Finally, the AI event in the gene was assigned if the pooled $P$-value was ≤0.05.

Finally, from 201 tested cancer genes, 64 genes with MMs in their canonical cancer types, 19 epigenetically silenced genes (>1% in their canonical cancer types; 30 pairs), and 159 genes with AI events, were analyzed. In total, 32 interactions between mutation and AI were detected (FDR 10%), including significantly overlapping 2-way interactions from CNA loss model (20 interactions; 62.5%, odds ratio = 10.23, $P = 2.2E-16$) (Supplementary Fig. 7a and Supplementary Data 6). From 64 cancer genes with MM events in TCGA, three genes (APC, PTEN, and PIK3CA) presented very high MMs frequencies in Colon/Rectum adenocarcinoma (COADREAD) and Uterine Corpus Endometrial Carcinoma (UCEC) that is assigned to the one-hit driver from CNA loss model (Supplementary Fig. 7c). In particular, APC in COADREAD has 38.1% of MMs compared to 67.9% of mutated samples (100% of MMs were in cis, i.e., multiple mutations to the same alleles), PTEN in UCEC has 22.3% of MMs compared to 57.6% of mutated samples (majority of MMs, 68.8% were in trans, i.e., multiple mutations to the different alleles), and PIK3CA in UCEC has 6.1% of MMs from 43.7% of mutated samples (majority of MMs, 61.5% were in cis). Although we tested a diverse range of alternative possibilities that could contribute to two-way interactions, most one-hit drivers from the mutation-CNAs model still did not have alternative two-way interactions beyond the co-occurrences between mutations and CNAs.

**Statistical evaluation of mutation—CNAs association.** To determine the significance of the co-occurrence of a pair of somatic mutation (all kinds of non-synonymous) and CNAs within a gene across cancer types, we generated a model using a log-linear regression using the MASS package (version 7.3.53.1) in R[40]. Two separate models depending on CNAs (either gain or loss) have been performed in each individual gene-tissue pair as follows:

$$glm(N \sim mut + CNA\ gain + mut : CNA\ gain, family = poisson\ (\text{link} = \text{"log"})) \tag{1}$$

$$glm(N \sim mut + CNA\ loss + mut : CNA\ loss, family = poisson\ (\text{link} = \text{"log"})) \tag{2}$$

where: $mut$ = number of samples with somatic mutation of gene A; CNA-gain (or loss) = number of samples with CNA gain or amplified (or loss) of gene A; $mut$:CNA-gain (or loss) = number of samples with both mutation and CNA gain or amplified (or loss) of gene A. Therefore, two-way interactions between mutation and CNA gain (or loss) were measured by comparing between CNA wild-type (no copy-number changes) and CNA loss (only one-copy loss) or CNA gain (one-copy gain or more than two-copy gain) samples. The regression coefficient and $P$-value were computed for individual gene-cancer type pairs and derived from the $mut$:CNA gain (or loss) using the summary function in R. More negative values from CNA loss model refers to stronger co-occurrence between mutation and CNA loss within a gene-tissue pair, whereas more positive value from the CNA gain model represents stronger co-occurrence between mutation and CNA gain within a gene-tissue pair. To determine the significance of the co-occurrence of a pair of two genomic events in the same gene, we applied a permutation strategy[41] that controls for the heterogeneity in genomic alterations within and across samples. Using the permatswap function in the R package vegan (version 2.5.7) (http://vegan.r-forge.r-project.org), we produced permuted genomic alteration matrices that maintain the total number of genomic alterations for each alteration across samples as well as the total number of alterations per sample. Somatic mutation, CNAs loss, and gain events were considered as separate classes and the permutation was performed for each cancer type separately. With 100 permutations, FDR is estimated as the ratio between the number of detected interactions in the permuted matrix (i.e., false interaction) and the number detected with the real data (i.e., true interaction) for each $P$-value cut-off (Supplementary Fig. 1).

The three-way model quantifies the strength of interactions of three genomic alterations from two genes with a background alteration of the second gene (gene B) mutation in a cancer type in a similar manner as the two-way interaction model within a gene (gene A). High-order three-way interactions were identified using the following models:

$$glm(N \sim mut + CNA\ gain + Target + mut : CNA\ gain + CNA\ gain : Target + mut : Target$$
$$+ mut : CNA\ gain : Target, family = poisson\ (link = \text{"log"})) \tag{3}$$

$$glm(N \sim mut + CNA\ loss + Target + mut : CNA\ loss + CNA\ loss : Target + mut : Target$$
$$+ mut : CNA\ loss : Target, family = poisson\ (link = \text{"log"})) \tag{4}$$

where: $mut$ = number of samples with somatic mutation of gene A; CNA-gain (or loss) = number of samples with CNA gain or amplified (or loss) of gene A; target = number of samples with somatic mutation of gene B; CNAs-gain (or loss):Target = number of samples with both somatic mutation of gene B and CNAs of gene A; $mut$:Target = number of samples with both somatic mutation of gene A and gene B; $mut$: gene A CNAs:Target = number of samples with somatic mutation of gene B when samples with both mutation and CNAs of gene A. 1 was added to each

frequency when running the regression model to avoid division by zero frequencies. From the equation, high-order three-way interaction regression coefficient and *P*-values are derived from the *mut:CNA gain (or loss):Target* with *summary* statistics in R.

To check the effect of different mutation types on the two-way interactions between mutation and CNAs, we tested three additional two-way models across (i) only premature truncation variants (PTVs), (ii) only predicted deleterious missense mutations (DelMis), and (iii) only non-deleterious missense mutations (ND_Mis). While several mutation type-specific 2-way interactions were identified (FDR 10%; four interactions from PTVs–CNA loss and two interactions from DelMis–CNA loss), the majority of two-way interactions across different mutations types were also detected when testing for interactions with all types of somatic mutations (original design). In detail, 90.7% of PTVs, 93.3% of DelMis, and 100% of NonMis were overlapped with all types of somatic mutations (FDR 10%) (Supplementary Fig. 8; Supplementary Data 7).

Next, we restricted our analysis to functional non-synonymous mutations and PTVs after removing putative neutral non-synonymous mutations. We first survey the frequencies of functional non-synonymous mutations (Func-NSY) *versus* putative neutral non-synonymous mutation (Neutral-NSY) by following the definition of Mina et al. [42]: Func-NSY when recurrently detected at the same amino acid position (i.e., hotspot mutations) or having evidence of their functional role and Neutral-NSY is considered all other non-synonymous mutations. We found that median frequencies for Func-NSY in our collected non-synonymous mutations are 66.7% across cancer types (median for TSGs = 42.6% and for OGs = 87.5%; Supplementary Fig. 9a). Next, we repeated a statistical test for two-way interactions between mutations and CNAs for all somatic mutations except for Neutral-NSY (that is, all PTVs + only Func-NSY) to evaluate the robustness of 2-way interactions when including putative Neutral-NSY or not. Overall, the Func-NSY only analysis presented very similar effect sizes to the original model (Pearson correlation coefficient (PCC) = 0.85 between coefficient values in an additional model and the original model in CNAs loss; PCC = 0.82 in CNAs gain). Also, the interactions identified using this Func-NSY only definition of somatic mutations strongly overlap with those identified in the original model at FDR 10%: 94.6% (53 over 56 interactions) in the CNA loss model and 85.7% (12 over 14 interactions) in the CNA gain model (Supplementary Fig. 9b). Furthermore, detected three-way interactions (FDR 10%) in the Func-NSY only model also strongly overlap with the original model: 55.5% (5 out of 9) overlapping in the CNA loss model and 60.0% (6 out of 10) in the CNA gain model (Supplementary Fig. 9c).

**Functional annotation**. Functional similarity between two genes with third-order interactions was tested for sharing functional annotation of GO biological process terms, molecular function terms were collected from DAVID 6.8 (https://david.ncifcrf.gov/)[43] and Reactome biological pathway (https://reactome.org/)[44].

**Essential genes from CRISPR–Cas9 and shRNA screening**. The 2134 common essential genes identified in a CRISPR–Cas9 screen, which show strong dependencies in >90% of pan-cancer cell lines, were downloaded from DepMap (https://depmap.org/portal/download/). Next, 297 conserved essential genes across three cancer types with 72 cell lines were collected from Marcotte et al. [45].

**Software for statistical analyses**. Regression models were performed in the R statistical programming environment (v.3.6.2). Libraries that are required include vegan v.2.5.7, MASS v.7.3.53.1, stringr v.1.4.0, reshape2 v.1.4.4, data.table v.1.14.0, dplyr v.1.0.5, viridis v.0.5.1. Figures were generated using ggplot2 v.3.3.3 and pheatmap v.1.0.12. All statistical analyses were carried out using Python 2.7(packages Stats v.1.2.1 and NumPy v.1.16.5).

**Reporting summary**. Further information on research design is available in the Nature Research Reporting Summary linked to this article.

## Data availability

This study re-analyzed published data sets, including tumor data sets from TCGA Pan-Cancer Atlas. The TCGA somatic mutation data (mc3.v0.2.8.PUBLIC.maf.gz) was downloaded from https://gdc.cancer.gov/about-data/publications/pancanatlas and copy-number alteration data were downloaded from Synapse (syn5049520). Gene Ontology (GO molecular function and biological process) was downloaded from the DAVID 6.8 (https://david.ncifcrf.gov/) and list of Reactome pathways was downloaded from the https://reactome.org/. A list of the somatic driver genes was compiled in Supplementary Table 1 from Bailey et al. (https://doi.org/10.1016/j.cell.2018.02.060). Epigenetic silencing data can be obtained by contacting the corresponding author of the original publication (https://doi.org/10.1016/j.celrep.2018.09.082). Multiple driver mutations (MMs) in cancer genes were obtained in Supplementary Table 3 from Saito et al. (https://doi.org/10.1038/s41586-020-2175-2), and allelic imbalance data can be obtained by contacting the corresponding author of the original publication (https://doi.org/10.1038/s41467-018-04900-7). A list of functional non-synonymous mutations was collected in Supplementary Table 1 from Mina et al. (https://doi.org/10.1038/s41588-020-0703-5). Source data are provided with this paper.

## Code availability

Source code for log-linear regression test is available at https://github.com/SolipParkLab/CancerFitness.

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

## Acknowledgements

We thank Luis Garcia-Jimeno for assistance with permutation. S.P. is supported by the Agencia Estatal de Investigación, Ministerio de Ciencia e Innovación (MCIN/AEI/ 10.13039/501100011033) through the RETOS project PID2019-109571RA-I00. This work was funded by the European Research Council (ERC) Starting grant (HYPER-INSIGHT, 757700) to F.S and ERC Consolidator (IR-DC, 616434) and Advanced (MUTANOMICS, 883742) grants to B.L. F.S. and B.L. are funded by the ICREA Research Professor program. S.P., F.S., and B.L. acknowledge the support of the Severo Ochoa Centres of Excellence program to the CNIO, IRB Barcelona, and to the CRG (MCIN/ AEI/10.13039/50110001103), respectively. B.L. and F.S. Work is funded with the grants BFU2017-89488-P and RegioMut BFU2017-89833-P (MCIN/AEI /10.13039/ 501100011033/FEDER "A way to make Europe"), respectively. B.L. is further supported by the Bettencourt Schueller Foundation, the Agencia de Gestio d'Ajuts Universitaris i de Recerca (2017 SGR 1322), and the Centres de Recerca de Catalunya (CERCA) program/ Generalitat de Catalunya. B.L. also acknowledges the support of the Spanish Ministry of Economy, Industry, and Competitiveness to the European Molecular Biology Laboratory (EMBL) partnership. The results shown here are in whole or part based upon data generated by the TCGA Research Network.

## Author contributions

S.P. performed all analyses. F.S. designed methods for testing interactions between mutation and CNAs. S.P., F.S., and B.L. designed analyses, evaluated the results and wrote the paper.

## Competing interests

The authors declare no competing interests.
