## [Peer review file · Nature Communications]

REVIEWER COMMENTS

Reviewer #1 (Remarks to the Author): Expert in cancer genomics and evolution

This is a manuscript in which the investigators use the recent TCGA data, >10K samples over >30 cancer types, to look at driver genes and their interactions in the context of mutations and copy number alterations (CNAs). They report context-dependent behavior, including so-called "third-order interactions" in which mutation-CNA dynamics in one gene are affected by mutation status in a different gene. It is a good topic in the sense that a significant effort within current cancer research is devoted to piecing-together the larger biochemical control systems relevant to cancer and understanding how they work under the insult of various genomic alterations. However, the finding/conclusion of this manuscript is not surprising or novel. I have a number of technical comments.

MAJOR COMMENTS

The authors use an FDR cutoff of 0.2 for most of their calculations. While 0.2 does appear in some papers, many investigators consider this value too permissive, in other words too high. What one finds in the literature are FDR cutoffs usually set in the range 0.01 (PMID 27259149) to 0.1 (PMID 27842059), with FDR of 0.05 being perhaps the most common value. The issue here is that some of significant findings seem to be in this nebulous region of FDR, which is to say some of the results depend strongly on how one picks FDR cutoff. For example, in Fig 2C, quite a few two-hit loss tissues would drop out for FDRs of 0.1 or 0.05 (respective y-axis values of 1 and 1.3) and almost all results would drop out for both two-hit loss and two-hit gain for FDR 0.01 (y-axis value of 2). The larger implication here, I think, is that statistical power may be borderline. On this same theme, a single gene (two-way) test from a pool of N genes is multiple-test-corrected for N tests, but when the so-called 3rd-order dynamics are examined (pairs of genes), the number of test corrections goes up to $N(N-1)/2$ (or thereabouts, depending upon whether "all vs all" is being examined, or just a subset). Overall, I think the multiple testing aspect requires more thorough treatment and explanation, including justifying the FDR threshold. I don't know if this was considered, but the log-linear analysis used here makes no distinction between independent and dependent variables. But the investigators are looking at the case where mutation in a 3rd-party gene is independent and can affect the mutation-CNA dynamics of a subject gene. Logistic regression is often used in such cases.

MINOR COMMENTS

In Line 51, the authors cite Knudson's classic paper, implying origination of the 2-hit hypothesis (many papers do this), but there were earlier contributions. Kern (PMID 12496492) relates the history of the discovery and it might be good to add this citation. There are a number of confusing issues related to figures. First, I do not understand the notation " $-(\text{Log}P, 10)$ " that appears in a number of figures. Do the authors mean " $-\text{Log}_{10}(P)$ "? If so, some bubble plots, e.g. in Fig 1, are littered with highly insignificant indicators. It may be good to leave only the significant ones in. That raises another observation. Plots mix P-values and FDRs, sometimes even within the same figure, e.g. Fig 2. This is very confusing. Since there are generally multiple tests, it would be most informative to stick with FDRs.

Reviewer #2 (Remarks to the Author): Expert in genetic interaction networks

Thank you for the opportunity to review "Higher order genetic interactions switch cancer genes from two-hit to one-hit drivers" by Park et al. This manuscript describes how genetic interactions modulate the dominance and dosage sensitivity of cancer drivers switching them between one-hit or two-hit drivers. Authors analyzed the TCGA pan-cancer data from 10,000 tumors focusing on somatic mutations and copy number aberrations datasets to compute the co-occurrence of mutations and copy

number changes within cancer driver genes. They then computed whether this co-occurrence was modified by the presence of a mutation in another gene. They find evidence that supports that the strength of the co-occurrence of mutation and copy number change in one gene is affected by the presence of a mutation in a second gene. Authors refer to this as a third order interaction between geneA-mutation, geneA-copy number change and geneB-mutation. They find that the two genes that exhibit such a genetic interaction are often functionally related.

This study sheds light on a previously unresolved finding in cancer datasets why are oncogenes amplified in one cancer but not others and why do tumor suppressors sometimes act as haploinsufficient one-hit drivers and in other cancers as recessive two-hit drivers. This study addresses an important question in the field of the cancer genetics and provides valuable and novel insight about how cancer mutations influence each other and the apparent discrepancy between cancer genes behaving as one-hit vs two-hit drivers. This is an elegant study and I had pleasure reading it.

Specific comments:

1. Is there a distinction between alleles when reporting co-occurrence between mutations and copy number changes i.e. do you know whether the mutated gene is also amplified or is a wild-type allele amplified?
2. Does the evolutionary trajectory influence whether you see one-hit vs two-hit mutations? Do two-hits mutations tend to happen as clonal events and one-hit as subclonal?
3. It is not clear what does Figure 2D show. '2-way' and '3-way' terms are used before authors define a third order genetic interaction in the next section. This needs to be clearly stated in the main text and the figure legend. Along the same lines, I don't understand what you mean by line 143-144: "For example, the interaction between BRAF mutation and CAN gain is stronger in SKCM than three other cancer significantly mutation cancer types (FDR<0.2)." Please, reword to clarify.
4. Generally, 'third order' genetic interactions refer to either 3 mutations within the same gene or 3 different genes. Referring to geneA-mutation, geneA-copy number change and geneB-mutation as a third order genetic interaction is unusual and needs to be defined very clearly on page 4 lines 154-156.
5. In the section 'Third order genetic interactions...', was the mutation in the second gene also annotated as loss of function or gain of function? Did it include copy number changes? Is the functional effect of the mutation in the second gene consistent with a LOF or GOF of the pathway in the context of the first mutation?
6. Page 5, lines 168-183, authors should elaborate on their observations and discuss the mutations as loss of function(KEAP) or gain of function(BRAF) phenotypes and the resulting interactions as sensitizing (PTEN) and suppressing (NRAS). Explicitly calling them as such will make the text more accessible for the reader.
7. In Figure 3H, Reactome pathway is misspelled 'pahway'.
8. In Figure 4, using different colours for the same gene is confusing, I would suggest using the same colour for mutations and copy number status of the same gene but perhaps use different shapes or even a cross to denote gene loss and multiple overlapping circles to denote gene gain.

Reviewer #3 (Remarks to the Author): Expert in cancer genomics and evolution

In this study, Park, Supek and Lehner look at single- and double-hits (CNA+mutation) in cancer genes and their third-order interactions with mutations in other genes across cancer types from the TCGA.

They collect somatic mutations status as well as information on gain/loss of material for cancer genes across TCGA. Then they use generalised linear modelling to infer interactions between mutations and presence of CNA loss/gain, as well as with somatic mutation in other genes.

They identify types of cancer genes with single hits, double-hits including either CNA gain or loss, as well as double hits including both gain and loss in the same cancer type.

They identify interactions with somatic events in other genes, and show these belong in general to the same pathway, suggesting double hits can be at a pathway- rather than gene-level.

Altogether, the questions and the hypotheses are relevant to the field and interesting. The paper is clearly written. But I believe the approach is not enough documented and the analyses, results and their discussion are lacking important details - please see comments.

Major comments.

The main methodology presented (i.e. the glm and R code formulas) is not clear to me: what is "N", is this never defined? if the glm are performed for each individual gene-tissue pair, what form do "mut" and "CNA gain" take, i.e. are these vectors of 0s/1s and absolute number of copies across samples? From the R code this is not clear either. Is the R code only for a single cancer type (KICH), so the glm are only run on a cancer type basis and never across cancer types? What is the value of the object "tumor"? A more complete description of the formula and possibly an example with (mock) data would help understand how the main questions are being formulated mathematically.

CNA events impacting cancer genes can be as large as whole-chromosomal event, think of chr17 losses in TP53 mutated cases, or very focused, such as chr7 EGFR amplifications in GBM. Moreover, CNA global profiles can also vary drastically from sample to sample, from diploid "flat" profiles to the whole genome being aberrated. In this context, those CNA events hitting given genes in highly aberrated samples would be much less specific than in diploid genomes. It is conceivable that samples with hits in important TSGs (double or single) such as TP53/RB1/etc., which might be more chromosomally aberrated, would in turn show with more (unspecific) double hits in other genes, which would be seen here as "third order interactions".

The interactions identified by the authors are not at all interpreted in that context, which I think is an important missing piece of information. And the rationale leading to the claim that 2 single-hits in independent genes from the same pathway indeed have similar effects to double-hits of the same gene is really not airtight.

Also, alternative epigenomic alterations are described in the methods, but it is not clear how they are used and represented in the main results. These other types of events can lead to silencing or amplifying genes, which should indeed be taken into account - but are they and how? Related to this, it is not clear to me how double-hits such as homozygous deletions are dealt with in the glm if at all. If they are not, it is unlikely that double hits (CNA + mutation) would be seen as single hits; however, the frequency of double hits in given genes might be significantly off.

In the methods, GISTIC is mentioned, which does not provide a copy-number estimate per se but rather a proxy for gain/loss of material which is relative to the average baseline log_r in the sample. It is unclear how the current definitions of the -2, 1, 0, 1, 2 GISTIC states relate to actual copy numbers. For example, in a whole-genome duplicated tumor (baseline 2+2 copies of each allele), how would a 3+0 vs. 2+1 or 1+1 vs. 2+0 be encoded? Clearly these states should be distinguished, as e.g. a 1+1+TP53mut vs. 2+0+TP53mut should represent single vs. double hits, respectively. This is true for copy-neutral loss of heterozygosity as well, which would be missed as a loss. Why not use actual copy-number values instead, which are available for TCGA?

Recurrent non-synonymous mutations in cancer genes are used here. But for each gene/cancer type, what is the expected proportion of those mutations that is not driver, i.e. the false positive rate? Using this proportion, for each gene, how many "single hits" would that induce across the cohort? How many third-order interactions would be inferred because of those, just by chance?

We thank the reviewers for their enthusiasm and insightful suggestions. Please see the point-by-point responses below for the changes that we have made in the revised manuscript.

Reviewer #1 (Remarks to the Author): Expert in cancer genomics and evolution

This is a manuscript in which the investigators use the recent TCGA data, >10K samples over >30 cancer types, to look at driver genes and their interactions in the context of mutations and copy number alterations (CNAs). They report context-dependent behavior, including so-called "third-order interactions" in which mutation-CNA dynamics in one gene are affected by mutation status in a different gene. It is a good topic in the sense that a significant effort within current cancer research is devoted to piecing-together the larger biochemical control systems relevant to cancer and understanding how they work under the insult of various genomic alterations. However, the finding/conclusion of this manuscript is not surprising or novel. I have a number of technical comments.

Thank you for the clear summary and enthusiastic evaluation.

MAJOR COMMENTS

1. The authors use an FDR cutoff of 0.2 for most of their calculations. While 0.2 does appear in some papers, many investigators consider this value too permissive, in other words too high. What one finds in the literature are FDR cutoffs usually set in the range 0.01 (PMID 27259149) to 0.1 (PMID 27842059), with FDR of 0.05 being perhaps the most common value. The issue here is that some of significant findings seem to be in this nebulous region of FDR, which is to say some of the results depend strongly on how one picks FDR cutoff. For example, in Fig 2C, quite a few two-hit loss tissues would drop out for FDRs of 0.1 or 0.05 (respective y-axis values of 1 and 1.3) and almost all results would drop out for both two-hit loss and two-hit gain for FDR 0.01 (y-axis value of 2). The larger implication here, I think, is that statistical power may be borderline. On this same theme, a single gene (two-way) test from a pool of N genes is multiple-test-corrected for N tests, but when the so-called 3rd-order dynamics are examined (pairs of genes), the number of test corrections goes up to $N(N-1)/2$ (or thereabouts, depending upon whether "all vs all" is being examined, or just a subset). Overall, I think the multiple testing aspect requires more thorough treatment and explanation, including justifying the FDR threshold. I don't know if this was considered, but the log-linear analysis used here makes no distinction between independent and dependent variables. But the investigators are looking at the case where mutation in a 3rd-party gene is independent and can affect the mutation-CNA dynamics of a subject gene. Logistic regression is often used in such cases.

The first issue that the reviewer raised is the use of permissive FDR thresholds (FDR 20%). We have updated our report to focus on a more stringent set of results at an FDR 10% (including cancer type-specific 2-way interactions and 3-way interactions). Moreover, we have updated the method to estimate false discovery rates (FDRs), which is now using a randomization strategy. To test the significance of the co-occurrence of a pair of two genomic events in the same gene, we used a previously described permutation strategy (Park *et al.*, 2015) that controls for the heterogeneity in genomic alterations within and across samples. Using the *permatswap* function in the R package *vegan*, we produced permuted genomic alteration matrices that maintain the total number of genomic alterations for each genomic alteration across samples as well as the total number of genomic alterations per sample. Somatic mutation, CNAs loss and gain events were considered as separate classes and the permutation was performed for each cancer type separately. With 100 permutations, FDR is estimated as the ratio between the number of detected interactions in the permuted matrix (i.e., false interaction) and the number detected with the real data (i.e., true interaction) for each p-value cut-off (**Revised Supplementary Figure 1**). We have also included an additional supplementary figure where we present the number of significant 2-way interactions detected between mutation and CNA-loss for

known tumor suppressor genes (TSGs) in their canonical tissues at different FDR thresholds (**Revised Supplementary Figure 2A**). At FDR=10%, we could detect 88.8% (56 out of 63 detected pairs) of known TSGs in their canonical cancer types. We also show the classification of cancer genes at different FDR cut offs (**Revised Supplementary Figure 2B**).

The second issue raised by the reviewer is about multiple testing correction: there is a concern that multiplicity of tests is very large due to a combinatorial nature of the testing, which incurs FDR penalties. However, it is not an “all vs all” testing in the strict sense: as the reviewer notes, we are selecting a subset of genes which are, in a particular cancer type, mutated at least 2% of the samples. For example, 19 genes total are considered in Colon/Rectum and 5 genes are considered in Ovarian cancer, meaning that even with $N(N-1)/2$ combinations the total number of tests is manageable (note that for the third-genomic alteration event, the number of genes does not increase because we consider only CNA in the pre-selected set of mutated genes). We would also like to note that the newly-implemented randomization test to estimate FDR – see above – lessens the penalties from having many correlated tests (this issue commonly makes the usual B-H correction, as we employed before, too conservative).

Finally, the reviewer touches upon our use of log-linear analysis (LLA; essentially, this is an application of count models [Poisson regression] to contingency tables). We understand that issue raised here is that LLA, in its ‘default’ usage, does not distinguish between dependant and independent variables, and so many different interactions are tested unnecessarily, thus increasing the FDR burden. While generally true, this does not apply to our case: we are not testing all three-way interactions theoretically testable by LLA, but only the subset of potential interactions, where the 3rd event (mutation in any genes) interacts with genes where there is already a 2-way interaction (between mutation and CNA in the first gene). Indeed, as the reviewer mentions we’re “looking at the case where mutation in a 3rd-party gene is independent and can affect the mutation-CNA dynamics of a subject gene”. It is currently not clear to us that applying logistic regression instead of LLA would bring benefits in terms of power or similar (particularly with the new FDR implementation above); if the reviewer thinks this is the case, we would be happy to do so.

MINOR COMMENTS

1. In Line 51, the authors cite Knudson's classic paper, implying origination of the 2-hit hypothesis (many papers do this), but there were earlier contributions. Kern (PMID 12496492) relates the history of the discovery and it might be good to add this citation.

We thank the reviewer for raising this point and added the suggested reference on Page 2.

2. There are a number of confusing issues related to figures. First, I do not understand the notation “-(LogP, 10)” that appears in a number of figures. Do the authors mean “-Log₁₀(P)”? If so, some bubble plots, e.g. in Fig 1, are littered with highly insignificant indicators. It may be good to leave only the significant ones in. That raises another observation. Plots mix P-values and FDRs, sometimes even within the same figure, e.g. Fig 2. This is very confusing. Since there are generally multiple tests, it would be most informative to stick with FDRs.

Figure 1 and 2 now present FDR values.

Reviewer #2 (Remarks to the Author): Expert in genetic interaction networks

Thank you for the opportunity to review “Higher order genetic interactions switch cancer genes from two-hit to one-hit drivers” by Park et al. This manuscript describes how genetic interactions modulate the dominance and dosage sensitivity of cancer drivers switching them between one-hit or two-hit drivers. Authors analyzed the TCGA pan-cancer data from 10,000 tumors focusing on somatic mutations and copy number aberrations datasets to compute the co-occurrence of mutations and copy number changes within cancer driver genes. They then computed whether this co-occurrence was

modified by the presence of a mutation in another gene. They find evidence that supports that the strength of the co-occurrence of mutation and copy number change in one gene is affected by the presence of a mutation in a second gene. Authors refer to this as a third order interaction between geneA-mutation, geneA-copy number change and geneB-mutation. They find that the two genes that exhibit such a genetic interaction are often functionally related.

This study sheds light on a previously unresolved finding in cancer datasets why are oncogenes amplified in one cancer but not others and why do tumor suppressors sometimes act as haploinsufficient one-hit drivers and in other cancers as recessive two-hit drivers. This study addresses an important question in the field of the cancer genetics and provides valuable and novel insight about how cancer mutations influence each other and the apparent discrepancy between cancer genes behaving as one-hit vs two-hit drivers. This is an elegant study and I had pleasure reading it.

We thank this reviewer for their positive remarks with regard to innovation.

Specific comments:

1. Is there a distinction between alleles when reporting co-occurrence between mutations and copy number changes i.e. do you know whether the mutated gene is also amplified or is a wild-type allele amplified?

Prompted by this reviewer suggestion, we investigated whether the mutated allele is amplified or wild-type allele is amplified by measuring mutant allele frequency (**Revised Supplementary Figure 5**). We assumed that if the mutated allele is amplified and therefore mutant allele frequencies (VAF) would be expected higher than 50%, whereas mutant allele frequencies would be expected lower than 50% if a wild-type allele is amplified. We compared mutant allele frequencies between samples with CNA gain and CNA wild-type as a control using 8 gene-tissue pairs which show significant co-occurrences between mutation and CNA gain (FDR = 10% with number of samples > 5 in both two sample groups). The majority of tested pairs (75.5%; 6/8 pairs) show significant increased VAFs in samples with CNA gain compared to CNA wild-type. These results were added to the results section on Page 4.

2. Does the evolutionary trajectory influence whether you see one-hit vs two-hit mutations? Do two-hits mutations tend to happen as clonal events and one-hit as subclonal?

This is an interesting question that we have explored in a preliminary analysis using the evolution histories of driver mutations (i.e., timing of point mutations in driver genes) for 2,658 tumors analyzed by the PCAWG consortium (including 653 TCGA samples) from Gerstung *et al*, 2020, Nature. From their analyses for the 50 most recurrent driver genes, 33 cancer genes overlapped with our study (from 73 cancer genes which are tested in at least two cancer types). To investigate differences in evolutionary histories between one-hit and two-hit drivers, we compared the odds ratio of clonal *versus* sub-clonal mutation (higher odds ratio: mutations are more enriched in clonal events than sub-clonal events). Two-hit drivers (either Class2 or Class3) are not significantly more enriched for clonal mutations than one-hit driver ($P = 0.17$; Mann-Whitney *U*-test; Reviewer Figure). We have not included this preliminary (negative) result in the revised manuscript because we think the question warrants more detailed investigation in the future.

3. It is not clear what does Figure 2D show. '2-way' and '3-way' terms are used before authors define a third order genetic interaction in the next section. This needs to be clearly stated in the main text and the figure legend.

'3-way' in Figure 2 has been changed to 'cancer type-specific 2-way' in the **revised Figure 2**.

Along the same lines, I don't understand what you mean by line 143-144: "For example, the interaction between BRAF mutation and CAN gain is stronger in SKCM than three other cancer significantly mutation cancer types (FDR<0.2)." Please, reword to clarify.

This has been reworded to: "*For example, the interaction between BRAF mutation and CNA gain in SKCM is stronger than in the three other cancer types in which BRAF is significantly mutated.*" in the revised manuscript page 5.

4. Generally, 'third order' genetic interactions refer to either 3 mutations within the same gene or 3 different genes. Referring to geneA-mutation, geneA-copy number change and geneB-mutation as a third order genetic interaction is unusual and needs to be defined very clearly on page 4 lines 154-156.

We have defined the type of 'third order' interaction that we are testing for on page 5.

We hypothesized that conceptually similar higher-order interactions might be occurring in cancer genomes, with mutations in a second gene (between gene interaction) altering the interaction between a mutation and CNA in a cancer gene (within gene interaction). Specifically, we tested for third order interactions involving two genetic alterations in one gene (somatic mutation and a copy number change) and a third alteration in a second gene (somatic mutation).

5. In the section 'Third order genetic interactions...', was the mutation in the second gene also annotated as loss of function or gain of function? Did it include copy number changes? Is the functional effect of the mutation in the second gene consistent with a LOF or GOF of the pathway in the context of the first mutation?

To identify third-order genetic interactions, we **only considered somatic mutations** (including all protein-truncation mutations, non-synonymous and missense substitutions) in the second gene. We did not consider copy-number changes in the second gene to avoid possible confounding by the overall level of copy number variation (see Reviewer 3 point #2 for the detail).

The question about pathway LOF vs GOF is an interesting question but we feel the limited number of examples we identify where mutations can be confidently classified as causing pathway activation or inactivation precludes a systematic analysis. However, anecdotally for the well-studied genes it appears to be the case, for example in the RTK/RAS pathway (*BRAF-NRAS* in SKCM) where the mutations in *BRAF* (the first gene) and *NRAS* (the second gene) contribute to activating RTK/RAS signalling pathway (revised **Figure 4**).

6. Page 5, lines 168-183, authors should elaborate on their observations and discuss the mutations as loss of function (KEAP) or gain of function (BRAF) phenotypes and the resulting interactions as sensitizing (PTEN) and suppressing (NRAS). Explicitly calling them as such will make the text more accessible for the reader.

We thank the reviewer for raising this point and have modified the text on page 5-6:

*Most of the third order interactions (76.5%, 13/17) are examples where the presence of a mutation in a second driver gene decreases the strength of the interaction between mutations and CNAs in the first driver (Fig.3C-F). For example, there is a strong interaction between mutations and CNA loss in KEAP1 in lung squamous cell carcinoma (LUSC) but not in samples that also carry a PTEN mutation. **This suggests that mutations in PTEN sensitize lung cells to the effects of reduced KEAP1 activity.** Similarly, in **Fig.3G** we show how the frequency of gene A mutations in samples carrying gene A CNAs varies depending up on the presence of mutations in gene B. For example, there is a strong third order interaction between BRAF mutations, BRAF CNA gain and NRAS mutations in skin cutaneous*

melanoma (SKCM). Only, 22% of samples with BRAF CNA gain and NRAS mutations have BRAF mutations whereas 82% of samples with BRAF CNA gain without NRAS mutations have BRAF mutations. This suggests that mutations in NRAS suppress the effects of increased BRAF activity, consistent with mutations in both genes activating the same pathway.

7. In Figure 3H, Reactome pathway is misspelled 'pahway'.

Corrected.

8. In Figure 4, using different colours for the same gene is confusing, I would suggest using the same colour for mutations and copy number status of the same gene but perhaps use different shapes or even a cross to denote gene loss and multiple overlapping circles to denote gene gain.

We sincerely appreciate this suggestion. We have applied the same colour to the same gene and different shapes to the different CNA classes as recommended by the reviewer in **revised Figure 4**.

Reviewer #3 (Remarks to the Author): Expert in cancer genomics and evolution

In this study, Park, Supek and Lehner look at single- and double-hits (CNA+mutation) in cancer genes and their third-order interactions with mutations in other genes across cancer types from the TCGA. They collect somatic mutations status as well as information on gain/loss of material for cancer genes across TCGA. Then they use generalised linear modelling to infer interactions between mutations and presence of CNA loss/gain, as well as with somatic mutation in other genes. They identify types of cancer genes with single hits, double-hits including either CNA gain or loss, as well as double hits including both gain and loss in the same cancer type. They identify interactions with somatic events in other genes, and show these belong in general to the same pathway, suggesting double hits can be at a pathway- rather than gene-level. Altogether, the questions and the hypotheses are relevant to the field and interesting. The paper is clearly written. But I believe the approach is not enough documented and the analyses, results and their discussion are lacking important details - please see comments.

We thank the reviewer for the excellent suggestions and enthusiastic evaluation.

Major comments.

1. The main methodology presented (i.e. the glm and R code formulas) is not clear to me: what is "N", is this never defined? if the glm are performed for each individual gene-tissue pair, what form do "mut" and "CNA gain" take, i.e. are these vectors of 0s/1s and absolute number of copies across samples? From the R code this is not clear either. Is the R code only for a single cancer type (KICH), so the glm are only run on a cancer type basis and never across cancer types? What is the value of the object "tumor"? A more complete description of the formula and possibly an example with (mock) data would help understand how the main questions are being formulated mathematically.

Thank you for pointing out this issue. In the revised manuscript (page 10) and code, we have clarified the description of the glm model. We first create a genomic alteration matrix across samples and cancer genes in a binary fashion (i.e., 0 indicates without alteration event, 1 indicates with alteration event) including three genomic alteration types (mutation, CNA loss, and CNA gain). Next, we counted the number of samples across four conditions, including (i) Mut_CNAs, number of samples with both mutation and CNAs, (ii) NoMut_CNAs, number of samples with only CNAs, without mutation, (iii) Mut_WT, number of samples with only mutation without CNAs, and (iv) NoMut_WT, number of samples neither mutation nor CNAs. In the R code formulas, N indicates the number of samples across four conditions, including "mut" presents the number of samples with mutation event and "CNA gain"

presents the number of samples with CNA gain event. The R code is designed to run for each cancer type separately (the object 'tumor' indicates a single cancer type that is tested).

2. CNA events impacting cancer genes can be as large as whole-chromosomal event, think of chr17 losses in TP53 mutated cases, or very focused, such as chr7 EGFR amplifications in GBM. Moreover, CNA global profiles can also vary drastically from sample to sample, from diploid "flat" profiles to the whole genome being aberrated. In this context, those CNA events hitting given genes in highly aberrated samples would be much less specific than in diploid genomes. It is conceivable that samples with hits in important TSGs (double or single) such as TP53/RB1/etc., which might be more chromosomally aberrated, would in turn show with more (unspecific) double hits in other genes, which would be seen here as "third order interactions".

The interactions identified by the authors are not at all interpreted in that context, which I think is an important missing piece of information. And the rationale leading to the claim that 2 single-hits in independent genes from the same pathway indeed have similar effects to double-hits of the same gene is really not airtight.

When testing for third-order genetic interactions we did **not** consider copy-number changes in the second genes – only somatic mutations. This is because, as the reviewer points out, associations between two CNAs could be non-specific due to large CNA events, aneuploidies etc. This is now more clearly stated in the main text on page 5.

3. Also, alternative epigenomic alterations are described in the methods, but it is not clear how they are used and represented in the main results. These other types of events can lead to silencing or amplifying genes, which should indeed be taken into account - but are they and how?

Thank you for this comment. As requested, we have added text to the main results to point out the alternative genomic alteration analyses on page 4. We tested for the alternative possibility of 2-way interactions through (i) promoter DNA hyper-methylation (silencing) and somatic mutation or (ii) promoter DNA hyper-methylation and CNAs loss. Using 35 epigenetically silenced genes-tissue pairs (> 1% in their canonical cancer types), only one significant 2-way interactions between hypermethylation and CNA loss—*ZNF133* in Ovarian – were identified (FDR 10%; **Revised Supplementary Figure 8**).

Related to this, it is not clear to me how double-hits such as homozygous deletions are dealt with in the glm if at all. If they are not, it is unlikely that double hits (CNA + mutation) would be seen as single hits; however, the frequency of double hits in given genes might be significantly off.

In the revised manuscript, only copy-number loss without homozygous deletions are considered in the CNA loss model. Therefore, 2-way interactions between mutation and CNA loss were measured by comparing between CNA WT (no copy number changes) and CNA loss (one copy loss) samples. This is now stated in the methods (page 10).

4. In the methods, GISTIC is mentioned, which does not provide a copy-number estimate per se but rather a proxy for gain/loss of material which is relative to the average baseline log_r in the sample. It is unclear how the current definitions of the -2, 1, 0, 1, 2 GISTIC states relate to actual copy numbers. For example, in a whole-genome duplicated tumor (baseline 2+2 copies of each allele), how would a 3+0 vs. 2+1 or 1+1 vs. 2+0 be encoded? Clearly these states should be distinguished, as e.g. a 1+1+TP53mut vs. 2+0+TP53mut should represent single vs. double hits, respectively. This is true for copy-neutral loss of heterozygosity as well, which would be missed as a loss. Why not use actual copy-number values instead, which are available for TCGA?

We thank the reviewer for pointing this out. Related to the above question (Question #3), we would like to clarify the definition of copy-number alterations in our study. In our model, samples with one-copy loss (-1 GISTIC states) were considered CNAs loss samples, whereas samples with one-copy gain (+1) or more than two-copy gains (high-level amplification; +2) were assigned to the CNA gain samples.

As the reviewer points out, we didn't consider the quantity of CNAs and allelic-specific mutation events in our model. We agree that addressing how interactions depend on the actual copy-numbers would be an important issue to address, but this is outside of the scope of the present study.

We tested for allelic imbalance (AI) to also consider copy-number neutral loss instead of copy-number reduction in our original study (**Revised Supplementary Figure 8**). 32 interactions between mutation and AI were detected (FDR 10%), including significantly overlapped 2-way interactions from CNA loss model (20 interactions; 62.5%, odds ratio = 10.23, $P = 2.2E-16$).

5. Recurrent non-synonymous mutations in cancer genes are used here. But for each gene/cancer type, what is the expected proportion of those mutations that is not driver, i.e. the false positive rate? Using this proportion, for each gene, how many "single hits" would that induce across the cohort? How many third-order interactions would be inferred because of those, just by chance?

We thank the reviewer for raising this point. To make it clear, somatic mutation calls were assigned to all protein-truncation mutations (PTVs) and to non-synonymous, single-residue substitutions in this study. As the reviewer pointed out, non-synonymous mutations could include non-driver (passenger) mutations. Prompted by the reviewer's suggestion, we first survey the frequencies of **functional non-synonymous mutations (Func-NSY)** versus **putative neutral non-synonymous mutation (Neutral-NSY)** by following the definition of Mina *et al.*, Nature Genetics, 2020: **Func-NSY** when recurrently detected the same amino acid position (i.e., hotspot mutations) or having evidence of their functional role and **Neutral-NSY is considered all other non-synonymous mutations**. We found that median frequencies for Func-NSY in our collected non-synonymous mutations are 66.7% across cancer types (median for tumor suppressor genes = 42.6% and for oncogenes = 87.5%; **Revised Supplementary Figure 7A**).

Next, we repeated the statistical test for 2-way interactions between mutations and CNAs for all somatic mutations except for Neutral-NSY (that is, all PTVs + only Func-NSY) to evaluate the robustness of 2-way interactions whether including putative Neutral-NSY or not. Overall, the new analysis presented very similar effect sizes to the original model (Pearson correlation coefficient (PCC) = 0.85 between coefficient values in a new model and the original model in CNAs loss; PCC = 0.82 in CNAs gain). Also, the interactions identified using this new definition of somatic mutations highly overlapped with the original model at FDR 10%: 94.6% (53 over 56 interactions) in CNA loss model and 85.7% (12 over 14 interactions) in CNA gain model (**revised Supplementary Figure 7B**). Furthermore, detected 3-way interactions (FDR 10%) in the new model are also highly overlapped with the original model: 55.5% (5 out of 9) overlapping in CNA loss model and 60.0% (6 out of 10) in CNA gain model (**revised Supplementary Figure 7C**).

In the original manuscript, we also presented interaction analyses for three different sets of somatic mutations: (1) only protein-truncation mutations (PTVs), (2) only predicted deleterious missense mutations (at least one of two tools (SIFT and PolyPhen2) predicted as deleterious/damaging variants; DelMis), and (3) only non-deleterious missense mutations (NonMis). While several mutation-type specific significant 2-way interactions have been identified (FDR 10%; 4 interactions from PTVs–CNA loss and 2 interactions from DelMis–CNA loss), majority of 2-way interactions across different mutations types were already found coincident with merged all types of somatic mutations (original design). In details, 90.7% of PTVs, 93.3% of DelMis, and 100% of NonMis were overlapped with all types of somatic mutations (FDR 10%) (**Revised Supplementary Figure 9**).

Reviewer Figure

Reviewer Figure. Relative timing of alterations in cancer genes, calculated as the odds ratio of clonal versus subclonal events from (Gerstung et al., 2020) between one-hit driver (class 1) and two-hit driver (class 2 or 3). Higher odds ratio indicate mutations are more enriched in clonal events than sub-clonal events.

REVIEWERS' COMMENTS

Reviewer #1 (Remarks to the Author):

The authors have addressed technical concerns raised. The novelty of the manuscript is limited.

Reviewer #2 (Remarks to the Author):

Authors addressed most of my comments sufficiently. I have no further suggestions, with the exception that in the revised Sup Fig 5A, authors appeared to have swapped the right most panels. The bottom panel should be on top and vice versa, otherwise if in the middle panel scenario a mutant allele is amplified, you would produce all amplified mutant alleles without any amplified WT alleles. This should be corrected. I recommend this manuscript for publication.

Reviewer #3 (Remarks to the Author):

I thank the authors for the care taken in responding to each of my comments.

REVIEWERS' COMMENTS

Reviewer #1 (Remarks to the Author):

The authors have addressed technical concerns raised. The novelty of the manuscript is limited.

We would like to thank the reviewer once again for your constructive questions and suggestions.

Reviewer #2 (Remarks to the Author):

Authors addressed most of my comments sufficiently. I have no further suggestions, with the exception that in the revised Sup Fig 5A, authors appeared to have swapped the right most panels. The bottom panel should be on top and vice versa, otherwise if in the middle panel scenario a mutant allele is amplified, you would produce all amplified mutant alleles without any amplified WT alleles. This should be corrected. I recommend this manuscript for publication.

We would like to thank the reviewer once again for your thoughtful questions and suggestions. This has now been corrected in the **Supplementary Figure 5A**.

Reviewer #3 (Remarks to the Author):

I thank the authors for the care taken in responding to each of my comments.

We would like to thank the reviewer once again for your positive comments.